# Immature excitatory neurons develop during adolescence in the human amygdala

Shawn F. Sorrells[1,2], Mercedes F. Paredes[3], Dmitry Velmeshev[1], Vicente Herranz-Pérez [4,5], Kadellyn Sandoval[3], Simone Mayer[1], Edward F. Chang[2], Ricardo Insausti[6], Arnold R. Kriegstein[1], John L. Rubenstein[7], Jose Manuel Garcia-Verdugo[4], Eric J. Huang [8] & Arturo Alvarez-Buylla[1,2]

The human amygdala grows during childhood, and its abnormal development is linked to mood disorders. The primate amygdala contains a large population of immature neurons in the paralaminar nuclei (PL), suggesting protracted development and possibly neurogenesis. Here we studied human PL development from embryonic stages to adulthood. The PL develops next to the caudal ganglionic eminence, which generates inhibitory interneurons, yet most PL neurons express excitatory markers. In children, most PL cells are immature (DCX+PSA-NCAM+), and during adolescence many transition into mature (TBR1+VGLUT2+) neurons. Immature PL neurons persist into old age, yet local progenitor proliferation sharply decreases in infants. Using single nuclei RNA sequencing, we identify the transcriptional profile of immature excitatory neurons in the human amygdala between 4–15 years. We conclude that the human PL contains excitatory neurons that remain immature for decades, a possible substrate for persistent plasticity at the interface of the hippocampus and amygdala.

[1] Eli and Edythe Broad Center of Regeneration Medicine and Stem Cell Research, University of California, San Francisco, San Francisco, CA 94143, USA. [2] Department of Neurological Surgery, University of California, San Francisco, San Francisco, CA 94143, USA. [3] Department of Neurology, University of California, San Francisco, San Francisco, CA 94143, USA. [4] Laboratory of Comparative Neurobiology, Institute Cavanilles, University of Valencia, CIBERNED, 46980 Valencia, Spain. [5] Predepartamental Unit of Medicine, Faculty of Health Sciences, Universitat Jaume I, 12071 Castelló de la Plana, Spain. [6] Human Neuroanatomy Laboratory, School of Medicine and CRIB, University of Castilla-La Mancha, 02006 Albacete, Spain. [7] Department of Psychiatry, Rock Hall, University of California, San Francisco, San Francisco, CA 94158-2324, USA. [8] Department of Pathology, University of California, San Francisco, San Francisco, CA 94143, USA. Correspondence and requests for materials should be addressed to A.A.-B. (email: AlvarezBuyllaA@ucsf.edu)

The adult birth of neurons occurs in the hippocampus subgranular zone (SGZ) and ventricular–subventricular zone (V–SVZ) in many mammals. Additional brain regions have been hypothesized to continue to produce new neurons in adulthood, including the amygdala, a center for fear and anxiety learning. Several studies report evidence of few rare new neurons in the amygdala of rodents[1,2] and primates[3] using the thymidine analog 5-bromodeoxyuridine (BrdU) to detect adult-born cells. It is not known whether the few reported adult-born neurons are adding to or replacing existing neurons, and in humans the presence of adult neural stem cells in the amygdala has not been demonstrated.

A region of the primate amygdala located near the temporal lobe lateral ventricle (tLV) called the paralaminar nuclei (PL), does, however, contain a large population of neurons expressing the microtubule-associated protein doublecortin (DCX) and polysialylated neural cell adhesion molecule (PSA-NCAM), both of which are found in immature neurons. This region contains cells with simple morphology and immature gene expression signature in several species of monkeys[3–7] and in humans[8,9]. Some of the immature neurons may be adult-born based on BrdU labeling in monkeys[3]. In humans, cell divisions have been detected in the adult temporal lobe V–SVZ[9], but immature migrating cells between the regions of proliferation and regions where immature neurons are found were not reported. Nevertheless, the PL appears to grow significantly in size and neuron number during juvenile ages in monkeys[4,7] and humans[10]. This growth could be due to increases in the size and complexity of post-mitotic cells within the amygdala, or it could result from the continued generation of new neurons from a pool of dividing neural stem cells. These increases in amygdala neurons are not seen in the brains of individuals with autism spectrum disorder (ASD)[10].

In the adult human amygdala the PL is located near the dorsal wall of the tLV, next to the caudal ganglionic eminence (CGE), a site of birth of GABAergic inhibitory interneurons during embryonic and fetal development that expresses the transcription factors SP8, PROX1, and COUP-TFII (Nr2f2)[11–14]. In rodents, dense clusters of GABAergic cells are located around the baso-lateral amygdala (BLA) and these clusters, called the intercalated nuclei, are similar in location and appearance to the PL, but do not express DCX and PSA-NCAM[15]. Interestingly, both the intercalated nuclei and the PL contain Bcl-2+ cells in adult non-human primates[16] and, like in rodents the intercalated nuclei in non-human primates contain GABAergic cells[17]. Based on these observations, it is possible that the PL and intercalated cells share similar origins and neuronal types. The GABAergic cells in the intercalated nuclei have been lineage-traced in rodents to the dorsal lateral ganglionic eminence (LGE)[18] and express transcription factor SP8, but the embryonic development and identity of the cells in the human PL has not been explored.

The presence of immature neurons with delayed maturation in adult brains has generated much interest as these cells could be substrates for plasticity and learning[19,20]. Neurons that retain the immature markers DCX and PSA-NCAM for prolonged periods of time, but were born earlier in development, have been found in the adult brain in a number of different species[20–24]. In adult mice, aside from the V–SVZ and SGZ, immature DCX+PSA-NCAM+ cells are restricted to layer II of the piriform cortex[20,25,26], and are not evident next to the amygdala. Animals with larger brains including primates, cats, whales, dolphins, pigs, and sheep have similar populations of immature neurons in the cortex, white matter, and PL[15,22,23].

The role of structural cellular plasticity in the PL is unknown, however, in primates this region is a primary target of hippocampal inputs[6]. The activity of the hippocampal–amygdala pathway is strongly associated with self-reported mood in human patients[27]. This places cellular plasticity within the PL at the interface between recent autobiographical memory in the hippocampus[28] and emotional valence processing by the amygdala[29,30]. In brain regions that continue to recruit new neurons like the olfactory bulb and dentate gyrus in mice, the addition of new cells over time has been suggested to facilitate the discrimination of similar inputs[31–33]. Interestingly, lesion studies of the macaque neonatal and adult uncal hippocampus have found fewer immature neurons and more mature neurons in the PL, implying a role for the hippocampal afferents on the maturation of these cells[34]. Early life stress also leads to lower levels of the excitatory neuron transcription factor T-box, brain 1 (TBR1) in the non-human primate PL[35]. A better understanding of the types of neurons in the PL, their relationship to adult neural stem cells, and their developmental dynamics is required to understand the implications of changes in neuropsychiatric disorders[10] and the relationship of the amygdala–hippocampal network to mood[27].

Here we investigated the identity of the immature cells in the human PL by following the development of this region from gestation into adulthood. We found that the PL can be identified based on expression of the excitatory neuron transcription factors TBR1 and COUP-TFII in mid-gestation (22 GW), much earlier than previously described. This revealed a clear distinction between the PL and the intercalated cells or CGE: unlike these regions, the PL contains mainly excitatory neurons. In addition, we found protracted maturation, and possible continued migration, of PL excitatory neurons that occurs rapidly during adolescence and continues throughout life with little evidence of proliferating progenitors specifically associated with this region.

## Results

**The amygdala PL form in gestation.** The PL develops next to the CGE which is highly proliferative in humans[36]. To determine whether the PL forms from progenitors in the CGE we immunostained the human temporal lobe for Ki-67 to label dividing cells, and for transcription factors expressed in the CGE (SP8, COUP-TFII, and PROX1)[13,36–38] (see Supplementary Tables 1 and 2 for complete list of cases and antibodies). At 22 gestational weeks, we found a region with a high density of Ki-67+ cells that extended anteriorly ventral to the developing amygdala around the ventricle (Fig. 1a–d, higher magnification in Fig. 1e), decreasing in density from posterior to anterior (Fig. 1b). There was a layer of Ki-67+SP8+ cells in the dorsal anterior temporal lobe ventricular wall wrapping around the tip of the ventricle (Fig. 1c, e). Regions with high Ki-67+ cell density contained many SP8+COUP-TFII+ cells, few NKX2.1+ cells that would be likely be medial ganglionic eminence-derived, and a dense field of vimentin+nestin+ cells and DCX+PSA-NCAM+ cells (Fig. 1f, g), suggesting that this anterior extension of Ki-67+ cells corresponds to the ventral anterior tip of the CGE. Staining for the transcription factors COUP-TFII, SP8, and PROX1 revealed widespread COUP-TFII expression in both the amygdala and CGE. The COUP-TFII+ cells in the CGE were SP8+PROX1+, whereas the COUP-TFII+ cells in the BLA and PL were primarily SP8−PROX1− (Fig. 1h, j–m). The developing PL was evident as a layer of higher-density COUP-TFII+SP8−PROX1− cells between the BLA and CGE, with a large medial region (MPL) and a thin lateral extension (LPL) (Fig. 1h, j–m, Supplementary Fig. 1). Similar COUP-TFII+SP8−PROX1− cells in the PL could be detected at birth, and 5 months (Fig. 1i). These observations suggest that the PL is distinct from the neighboring CGE.

To further characterize the identity of developing neurons in the PL we stained this region for the calcium-binding protein,

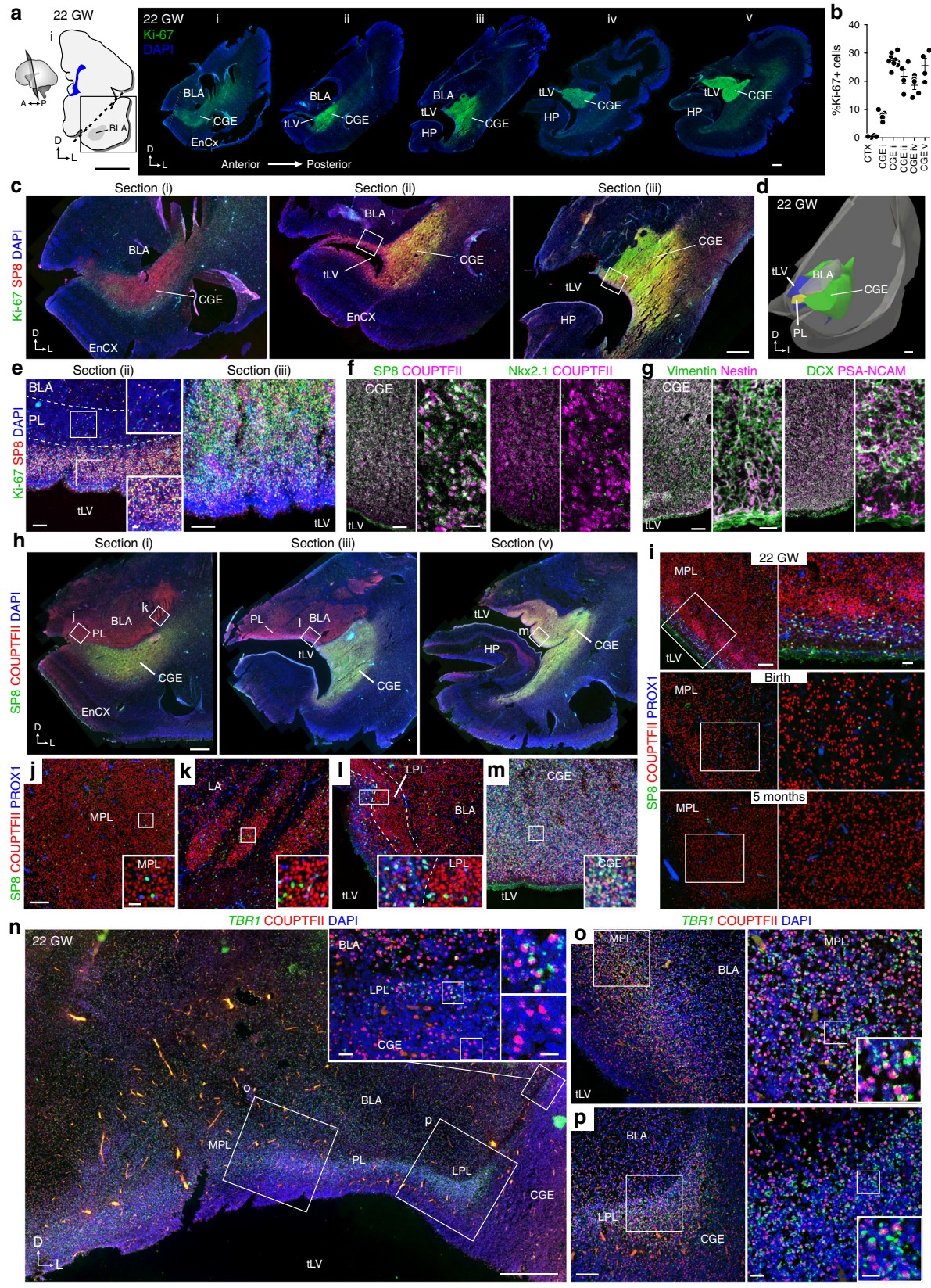

secretagogin (SCGN), expressed in CGE migratory inhibitory interneurons[39]. As expected, the CGE was densely populated with SCGN+ neurons, but there were few SGCN+ cells in the PL (Supplementary Fig. 1). Because these cells were negative for a marker of inhibitory neurons and distinct from the CGE,

we asked whether PL cells express the excitatory neuron transcription factor *TBR1*[40]. We used single-molecule in situ hybridization with *TBR1* human-specific probes to see if COUP-TFII+ cells in the PL expressed *TBR1*. At 22 GW, the dense COUP-TFII+ cells in the PL expressed *TBR1* mRNA, but

**Fig. 1** The PL is adjacent to the CGE at 22 gestational weeks (GW). **a** (Left) 22 GW human brain, coronal section of the basolateral amygdala (BLA). (Right) Ki-67+ cells in temporal lobe sections (i–v) spanning the anterior BLA to the caudal ganglionic eminence (CGE), including the entorhinal cortex (EnCX), and hippocampus (HP). **b** Percentage of Ki-67+ DAPI-stained nuclei in the temporal lobe cortex (CTX) or in the CGE in each section from **a**. (Left to right, n = 4, 11, 10, 4, 5, and 4 independent z-stacks analyzed per section, error bars s.e.m.). **c** Ki-67+SP8+ cells beneath the amygdala and surrounding the anterior tip of the temporal horn of the lateral ventricle (tLV) in sections i–iii from **a**. **d** 3-D reconstruction of sections in **a** showing the CGE extending beneath the BLA and the PL. **e** Boxed regions in sections ii and iii from **c** showing (Left) few Ki-67+SP8+ cells in the PL, more between the PL and ventricle, and (Right) many in the CGE. **f** SP8+COUP-TFII+ cells and few NKX2.1+ cells in the CGE at 22 GW. **g** Vimentin+nestin+ and DCX+PSA-NCAM+ cells in the CGE at 22 GW. **h** (Top) Coronal sections adjacent to i, iii, and v from **a** showing COUP-TFII+SP8− cells in the BLA, and COUP-TFII+SP8+ cells in the CGE. **i** COUP-TFII+SP8−PROX1− cells in the MPL at 22 GW, birth, and 5 months postnatal. **j–m** Higher magnification of boxed regions in **h**. COUP-TFII+ SP8−PROX1− cells in the (**j**) medial PL (MPL), (**k**) lateral amygdala (LA), and (**l**) lateral PL (LPL) next to (**m**) COUP-TFII+SP8+PROX1+ cells in the CGE. **n–p** COUP-TFII+ cells in the (**o**) MPL and (**p**) LPL express *TBR1* mRNA, unlike the COUP-TFII+ cells in the adjacent CGE (inset). Scale bars: 10 mm (a left), 1 mm (**a** right, **c**, **d**, **h**), 500 μm (**n**), 100 μm (**e**, **f** left, **i** left, **j–m**, **o** left, **p** left), 20 μm (**e** inset, **f** right, **g** right, **i** right, **j–m** insets, **n** left inset, **o** right, **p** right), 10 μm (**n–p** right insets)

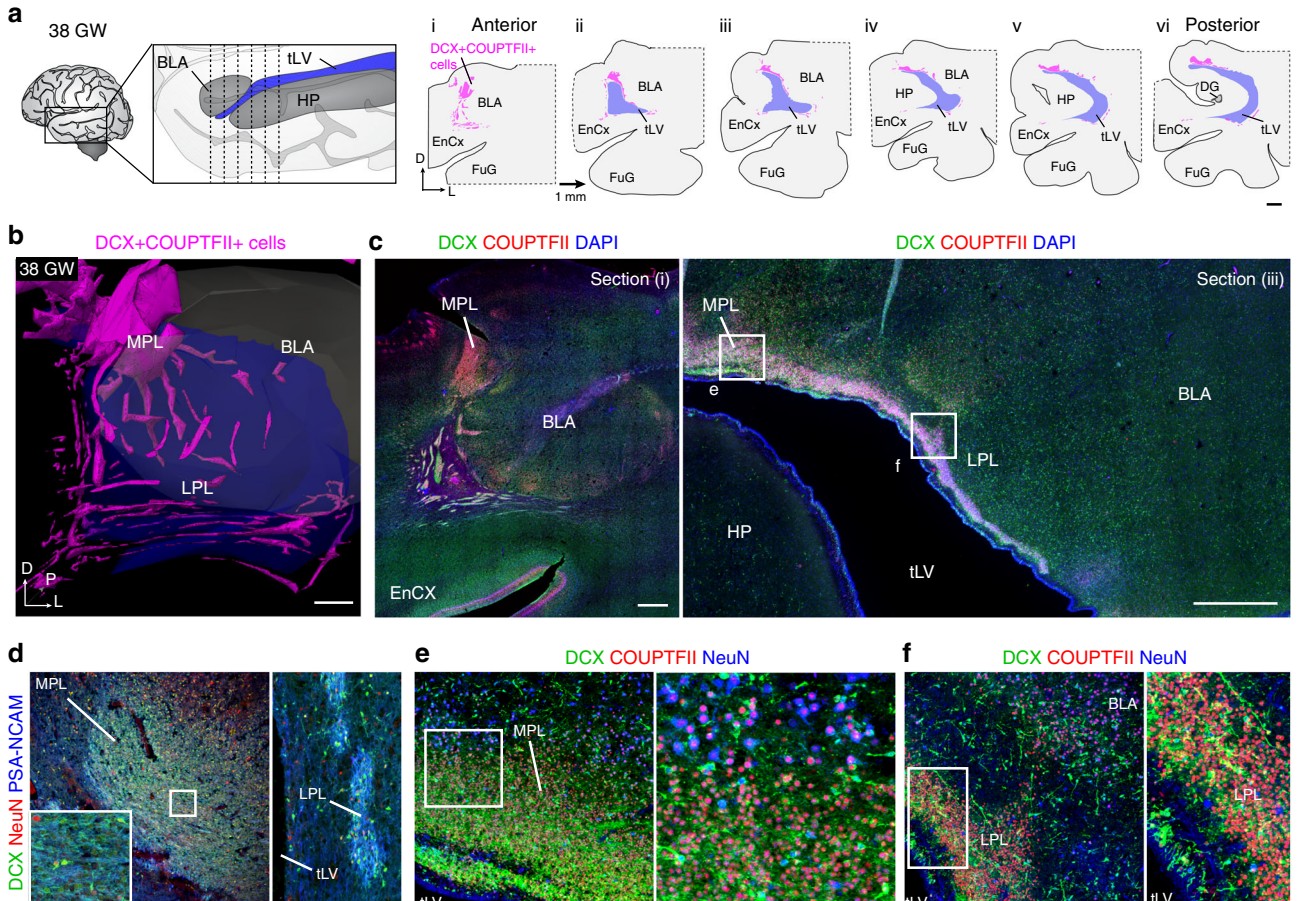

**Fig. 2** Distribution of DCX+COUP-TFII+ cells in the PL at birth. **a** Schematic of 38 GW temporal lobe indicating approximate level of coronal sections separated by 1 mm. Maps of the locations of dense clusters of DCX+COUP-TFII+ cells on the medial and ventral sides of the amygdala beginning at the anterior tip of the temporal lateral ventricle (tLV) corresponding to coronal section (i). **b** 3-D reconstruction of DCX+COUP-TFII+ clusters (purple) around and within the BLA (gray). **c** DCX+COUP-TFII+ cells in the PL at anterior and posterior levels (i and iii). **d** DCX+PSA-NCAM+ cell clusters are mostly NeuN− in the MPL and LPL at birth. **e**, **f** DCX+COUP-TFII+NeuN− cells in the MPL and LPL at birth from sections adjacent to the boxed areas in (**c**, iii). Fusiform gyrus (FuG). Scale bars: 2 mm (**a**), 1 mm (**b**, **c**), 200 μm (**d** left), 100 μm (**e** left, **f** left), 50 μm (**d** left inset, **d** right, **e** right, **f** right)

the COUP-TFII+ cells in the adjacent CGE were *TBR1* negative (Fig. 1n–p).

We next asked if the PL contained similar cellular identities at older ages by studying the ventral amygdala in late gestation and early postnatal life. Between 38 GW and birth, dense DCX+COUP-TFII+ cells were present in clusters along the entire anterior–posterior axis of the amygdala (Fig. 2a–d). These dense cell clusters were filled with DCX+ cells that were weakly TBR1+ but the

majority of these DCX+ cells expressed *TBR1* mRNA (Supplementary Fig. 1). At birth, the PL could be distinguished from the rest of the amygdala (dorsally), and the remnants of the CGE (ventrally), by the higher cell density and increased number of DCX+ and PSA-NCAM+ cells (Supplementary Fig. 2). At this age there were few NeuN+ neurons or OLIG2+ cells in the PL, and GFAP+ and SOX2+ cells were primarily located around the dense clusters of cells in PL (Supplementary Fig. 2). Together these data show that the PL is

distinct from the CGE and the rest of the amygdala, and contains immature *TBR1*-expressing excitatory neurons.

**Excitatory neurons slowly mature in the PL throughout life**. We next followed the DCX$^+$PSA-NCAM$^+$ cells in the PL later into childhood, adolescence, and adulthood. Mapping the locations of the DCX$^+$PSA-NCAM$^+$ cell clusters revealed a decrease in the number of DCX$^+$PSA-NCAM$^+$ cell clusters in the PL (mean 84 clusters per section (±32 clusters, s.d.) at birth (CI: 64–103; $n = 13$ sections, four individuals), compared to 9 clusters (±3 clusters, s.d.) in adults 24–77 years (CI: 7–11; $n = 17$ sections, four individuals); one-way ANOVA comparing birth ($n = 4$ individuals), birth–1 year ($n = 4$ individuals), 2–19 ($n = 5$ individuals) years, and 24–77 years ($n = 4$ individuals), $F (3, 58) = 58.91$, $p < 0.0001$) and an increase in the median area of each cluster (4508 μm$^2$ (±22,491 μm$^2$, s.d.) at birth (25th percentile: 2165 μm$^2$; 75th percentile: 9676 μm$^2$, $n = 1089$ clusters) compared to 51,041 μm$^2$ (±87,586 μm$^2$, s.d.) in adults 24–77 years (25th percentile: 16,655 μm$^2$; 75th percentile: 102,532 μm$^2$, $n = 160$ clusters); one-way ANOVA comparing areas for the same groups, $F (3, 1774) = 155.4$, $p < 0.0001$) (Fig. 3a, b). At birth there were few NeuN$^+$ cells in or adjacent to the dense DCX$^+$PSA-NCAM$^+$ cell clusters, but by 6 and 13 years both the MPL and LPL contained many large NeuN$^+$ neurons. Interestingly, some of the large NeuN$^+$ neurons were also DCX$^+$PSA-NCAM$^+$, and some of the small DCX$^+$PSA-NCAM$^+$ cells were weakly NeuN$^+$ (Fig. 3c, d). These data suggest that during childhood the small DCX$^+$PSA-NCAM$^+$NeuN$^-$ cells mature into DCX$^+$PSA-NCAM$^+$NeuN$^+$, and, into large DCX$^-$PSA-NCAM$^-$NeuN$^+$ neurons.

We next quantified the number of (DAPI+) nuclei, DCX$^+$PSA-NCAM$^+$ cells, and mature NeuN$^+$ neurons in the MPL and LPL from birth to 77 years. Prior to 2 years of age, DCX$^+$PSA-NCAM$^+$ cells in the MPL and LPL were densely packed with few NeuN$^+$ neurons between them (Fig. 4a, b). With increasing age, the density of the DAPI$^+$ cells in the PL decreased (Fig. 4c), the density of DCX$^+$PSA-NCAM$^+$ cells in the PL decreased (Fig. 4d), and the percentage of DAPI$^+$ cells that were DCX$^+$PSA-NCAM$^+$ cells also decreased (Fig. 4e). Concurrently, there was a significant increase with age in the density and percentage of NeuN$^+$ cells in the PL (Fig. 4f, g). The changes in DCX+PSA-NCAM+ and NeuN+ cells occurred significantly earlier in the MPL than the LPL (for statistical results see Supplementary Data 1). The above results suggest that immature neurons in the PL are slowly converting into mature neurons and this process is most pronounced during adolescence.

To examine possible connections between the PL and surrounding axonal tracts we made a 3-D reconstruction of DCX+PSA-NCAM+ cells within and surrounding the amygdala at 15 years of age (Supplementary Fig. 3). This revealed that the PL nuclei wrap around and extend into the basal and lateral amygdala nuclei at multiple locations along their entire anterior–posterior length. Interestingly, the extensions of the PL into the amygdala were wrapped with neurofilament+ fibers that extended from the angular bundle, a large fiber tract ventral to the amygdala. These neurofilaments came into very close proximity to the DCX+ cells in the PL, which themselves expressed the dendritic marker MAP2 (Supplementary Fig. 3).

We next investigated the ultrastructure of the cell clusters in the PL in an adult at 45 years of age. Using immunogold labeling we identified clusters of DCX+ neurons (Supplementary Fig. 3) with small nuclei filled with compacted heterochromatin, little cytosol, and few organelles. We observed cell–cell adhesion junctions between these cells and adjacent cells with larger nuclei, less heterochromatin, and more cytosol. These clusters also contained neurons with even larger nuclei, more cytosol, and

organelles, but which still had immature features. Together these observations confirmed that the PL, even in adults, contains neurons with ultrastructural properties indicative of different stages of maturation.

To characterize the identity of developing neurons in the PL we looked for other neuron markers that increased in this region with age. The percentage of large, calbindin (CB$^+$) neurons in the PL significantly increased between birth and 77 years (Fig. 5a–f, Supplementary Data 1). We next asked whether these large CB$^+$ cells expressed markers of excitatory neurons and found that they represented a subpopulation of the TBR1$^+$VGLUT2$^+$ cells in the MPL and LPL at 13 years of age (Fig. 5g–j). At 13 years we observed TBR1$^+$ small DCX$^+$ cells, TBR1$^+$ intermediate-sized DCX$^+$ cells, and large TBR1$^+$ NeuN$^+$ neurons (Fig. 5h, i). Of the small DCX$^+$ cells, 54.4% (±3.4%, s.d.; CI: 50.22–58.56; 972 TBR1$^+$DCX$^+$/1762 DCX$^+$ cells; $n = 5$ z-stacks) were TBR1$^+$ and of the large DCX$^-$TBR1$^+$NeuN$^+$ neurons, 97.6% (±3.5%, s.d.; CI: 93.32–100; 543 TBR1$^+$VGLUT2$^+$/564 TBR1$^+$ cells; $n = 5$ z-stacks) were VGLUT2$^+$ (Fig. 5i, j). We also looked for inhibitory interneurons in the PL and found a small population (~3% of DAPI$^+$ nuclei) of GAD67-expressing cells in the MPL and LPL at all ages; their number did not appear to change with age (Supplementary Fig. 4). Small subpopulations of cells in the PL-expressing interneuron markers calretinin (CR), neuronal nitric oxide synthase (nNOS), neuropeptide Y (NPY), or somatostatin (SST) also did not appear to change with age (Supplementary Fig. 4). Together these data suggest that the majority of large NeuN$^+$ cells in the PL are excitatory (TBR1$^+$VGLUT2$^+$) neurons, and their number increases postnatally and throughout adulthood in what appears to be a very protracted process of maturation.

**Dividing precursors decline in the first two years of life**. Due to the persistence of DCX$^+$PSA-NCAM$^+$ cells in the PL, we next asked if this region contains populations of dividing neural precursor cells. At birth there were Ki-67$^+$ cells in the PL, but these cells were more frequent between the PL and the ventricle in the remnants of CGE (Supplementary Fig. 5). At 5 months, 6 years, and 24 years of age (Fig. 6a) there continued to be Ki-67$^+$ cells adjacent to, but rarely within, the PL. The density of Ki-67$^+$ cells in the PL dropped from 85.6 Ki-67$^+$ cells/mm$^2$ (±86.8 cells/mm$^2$, s.d.; CI: 50.56–120.7; 72 Ki-67$^+$/11,628 DAPI$^+$ cells; $n = 26$ z-stacks) at birth to 18.24 Ki-67$^+$ cells/mm$^2$ (±16.19 cells/mm$^2$, s.d.; CI: 9.36–27.13; 20 Ki-67$^+$/9043 DAPI$^+$ cells; $n = 29$ z-stacks) at 5 months of age. By 2 years, the density in the PL was 6.7 Ki-67$^+$ cells/mm$^2$ (±9.0 s.d. cells/mm$^2$; CI: 2.634–10.82; 7 Ki-67$^+$/2472 DAPI$^+$ cells; $n = 21$ z-stacks), similar to 5.1 Ki-67$^+$ cells/mm$^2$ (±11.2, s.d. cells/mm$^2$; CI: 0.459–9.693; 6 Ki-67$^+$/5377 DAPI$^+$ cells; $n = 25$ z-stacks) in the 24-year-old adult. Ki-67$^+$ cells within or around the PL did not overlap with the DCX$^+$PSA-NCAM$^+$ cells at any postnatal ages examined (Supplementary Fig. 5). At birth there were Ki-67$^+$BLBP$^+$ and Ki-67$^+$vimentin$^+$ cells in both the MPL and LPL and after 5 months of age the Ki-67$^+$ cells were BLBP$^-$ or vimentin$^-$ (Supplementary Fig. 6). Interestingly, we found some BLBP$^+$ or vimentin$^+$ cells with long radial processes within and surrounding the PL up to 6 years of life, but these cells were not Ki-67$^+$. These data suggested that the PL and surrounding tissue contain dividing precursor cells at birth that decline within the first 5 months of life; however, whether these cells are precursors of neurons, glial cells, or both was not clear.

We next asked whether the Ki-67$^+$ dividing cells in or near the PL expressed progenitor cell markers. At birth we also found many Ki-67$^+$ cells near the PL that expressed SOX2, which is present in neural stem cells, astrocytes, and oligodendrocyte progenitors[41–43]. These Ki-67$^+$SOX2$^+$ cells were detectable within the MPL and LPL, and were most frequently observed in the adjacent white

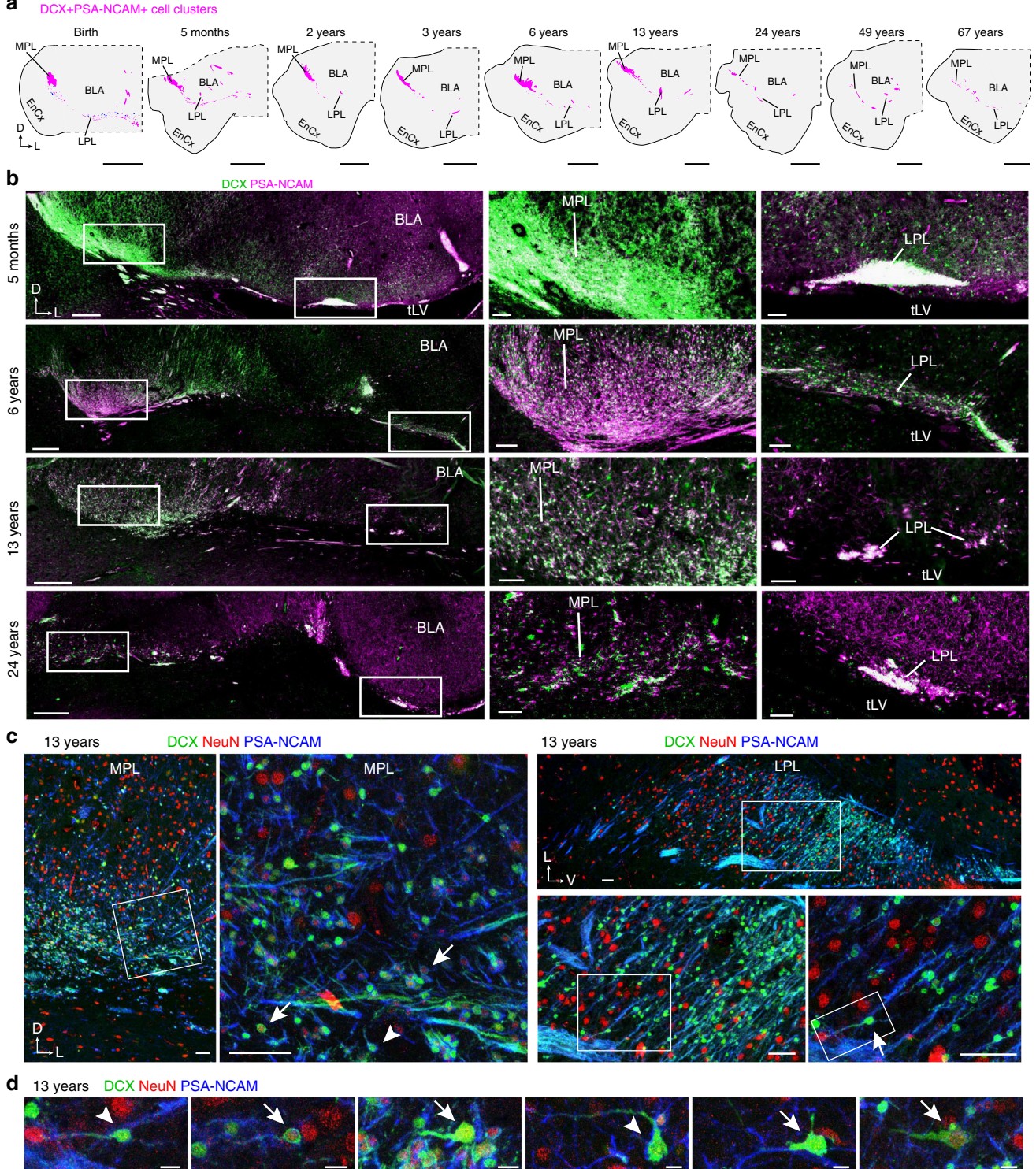

**Fig. 3** Postnatal maturation of DCX+PSA-NCAM+ neurons in the PL. **a** Maps of the location of DCX+PSA-NCAM+ cells in the PL from birth to 67 years of age, at approximately the same rostrocaudal level. **b** DCX+PSA-NCAM+ cells in the PL from 5 months to 24 years in coronal sections. (Right) Higher magnification of DCX+PSA-NCAM+ cells in the MPL and LPL. **c** DCX+PSA-NCAM+NeuN+ neurons in the 13-year-old MPL (Left) and LPL (Right). Some cells had the shape of migratory neurons (arrow). **d** Examples of DCX+PSA-NCAM+ cells that are NeuN+ (arrows) and NeuN− (arrowheads) in the MPL at 13 years of age. Scale bars: 5 mm (**a**), 500 μm (**b** left), 100 μm (**b** middle and right), 50 μm (**c**), 10 μm (**d**)

matter (Fig. 6b). By 5 months of age, Ki-67+SOX2+ cells had greatly decreased in number (Fig. 6c). At 2, 6, 13, and 24 years we could detect Ki-67+SOX2+ cells next to the DCX+ cells in the PL (Fig. 6d–f). Although there was a significant decrease with age, the density of the cells was similar in the MPL, LPL, and BLA

(Fig. 6e, f, Supplementary Data 1, 3). At birth we also found many Ki-67+ cells in the PL that expressed OLIG2 (36.3 ± 19.7%, s.d.; 277 Olig2+Ki-67+ cells/764 Ki-67+ cells, $n = 16$ z-stacks), which is present in oligodendrocyte progenitors. These cells were abundant in the white matter adjacent to the MPL and LPL

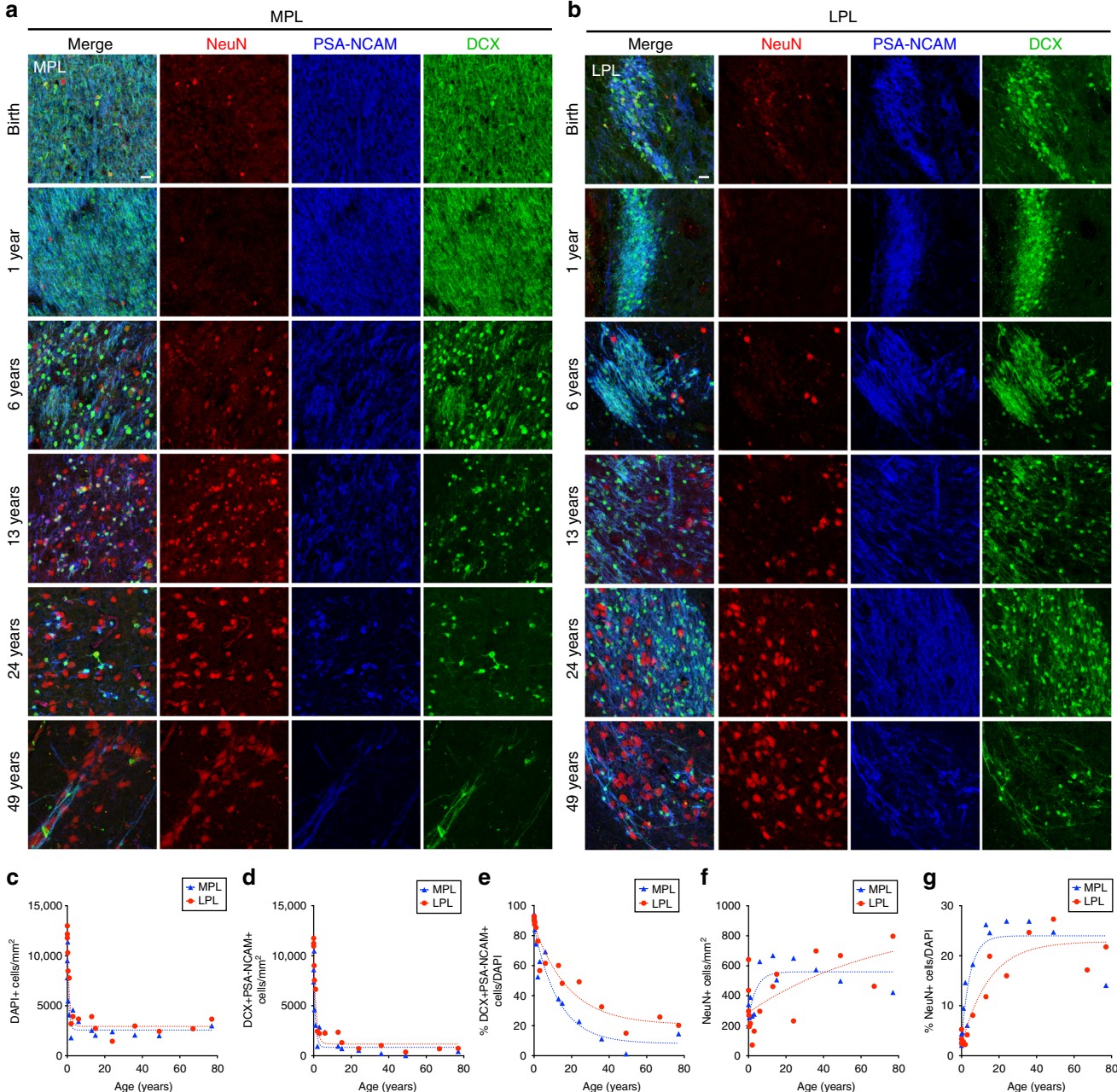

**Fig. 4** DCX+PSA-NCAM+ neurons decrease and NeuN+ neurons increase in the PL across life. **a**, **b** DCX+PSA-NCAM+ cells and NeuN+ cells in the MPL (**a**) and LPL (**b**) from birth to 24 years of age. **c–g** Quantification of cells in the MPL and LPL across life. Each data point corresponds to one individual; line indicates non-linear exponential fit. **c** Density of DAPI+ nuclei in the PL from birth to 77 years. **d** Density of DCX+PSA-NCAM+DAPI+ cells in the PL from birth to 77 years. **e** Percentage of DCX+PSA-NCAM+DAPI+ cells in the PL from birth to 77 years. **f** Density of NeuN+ cells in the PL from birth to 67 years. **g** Percentage of NeuN+ DAPI+ cells in the PL from birth to 67 years. Scale bars: 20 μm (**a**, **b**)

clusters, and were occasionally within the PL (Supplementary Fig. 7). By 5 months of age there was a sharp drop in the number of Ki-67+ cells in the PL and fewer of the remaining Ki-67+ cells were OLIG2+ (17.5 ± 18.7%, s.d.; 31 OLIG2+Ki-67+ cells/177 Ki-67+ cells, n = 23 z-stacks) between 5 months and 24 years (Supplementary Fig. 7). These data indicate a rapid decrease during early postnatal life in the number of Ki-67+ cells in or near the PL that express SOX2 or OLIG2.

**DCX+ cells with migratory morphology are present in adults.** The above data indicate that putative progenitor cells for new neurons close or within the PL are largely depleted within the first year of life; however, distant progenitors could continue to supply

the PL with immature neurons. DCX+ cells with a single long process and sometimes elongated nuclei were present in the PL at all ages studied, from 10 days to 77 years (Fig. 7a, b). To investigate the relationship between migratory neurons and the PL we mapped the location of DCX+ cells with a single leading process and elongated morphology from birth to 24 years of age (Fig. 7c). Before 1 year, cells with migratory morphology were also abundant throughout the human temporal lobe, amygdala, and PL. Their leading process was randomly oriented, with similar numbers of cells oriented toward the PL and away (Fig. 7c). After 2 years, the putative migratory cells were only observed within or in close proximity to the PL. If DCX+PSA-NCAM+ cells were migrating from distant locations into the PL, we would expect to

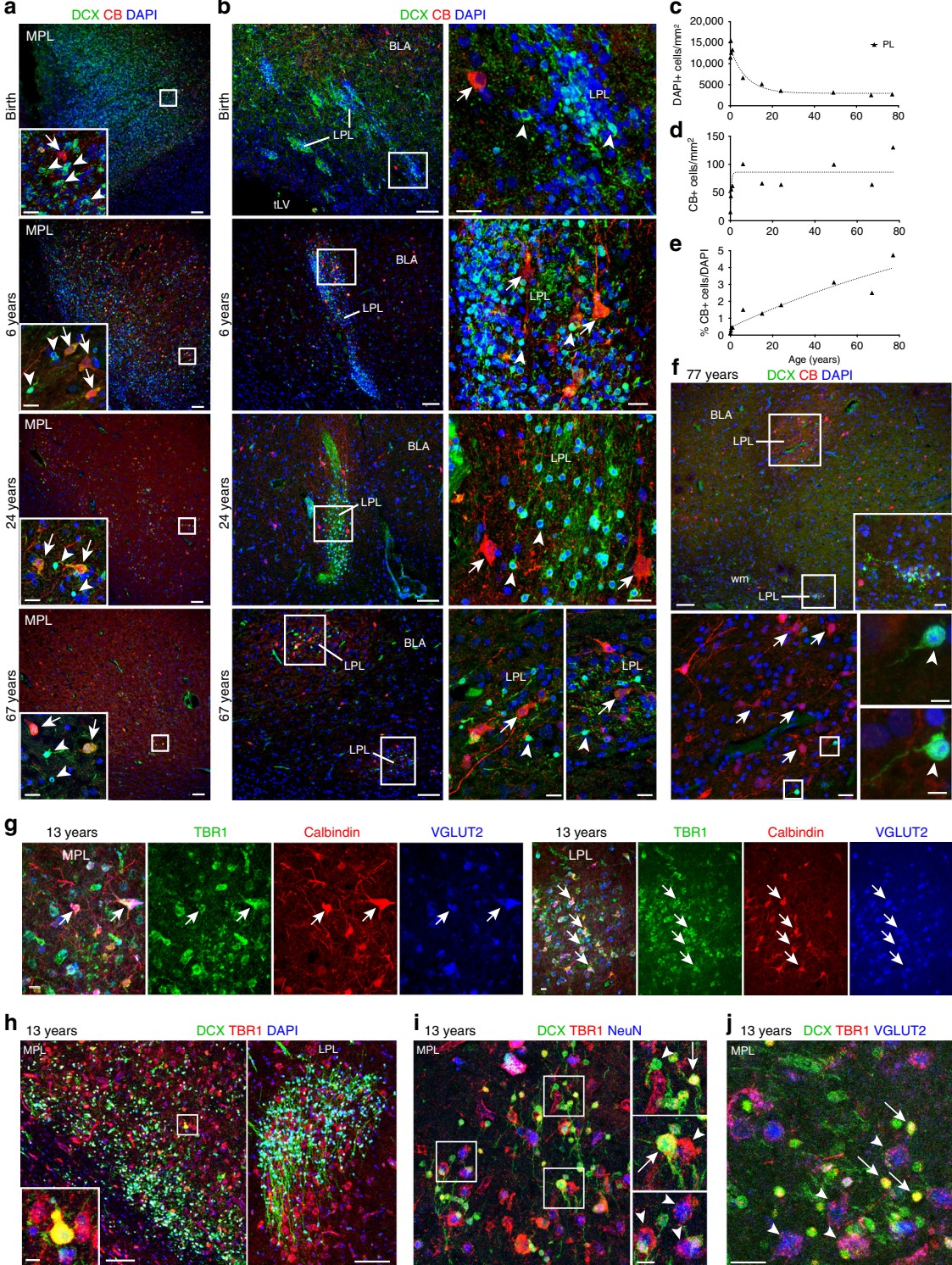

observe migrating neurons between the PL and the putative distant birthplace, but we did not observe this in young or old ages in coronal or sagittal sections. These observations suggest that widespread migration of young neurons occurs throughout the temporal lobe in children 5 months and younger, and after 2 years of age and into adulthood, cells with migratory morphology persist in the PL.

To gain additional insight into the nature of the immature PL neurons we used single-nucleus RNA sequencing (snRNA-seq) of eight flash frozen post-mortem human amygdala samples from 4 to 15 years of age from control and children with ASD (Supplementary Data 2). We isolated nuclei from the amygdala and profiled their transcriptomes with unbiased snRNA-seq using 10x Genomics platform. After filtering low-quality nuclei (see the section "Methods"), we generated 13,289 nuclei profiles. By performing clustering of single-nucleus transcriptional profiles (Fig. 8a, Supplementary Data 2), we identified a small but highly distinct cluster of cells (named cluster 2, C2) expressing DCX, BCL2, as well as NR2F2 (COUP-TFII) and a marker of immature neurons ROBO1 (Fig. 8b). Other clusters were associated with

**Fig. 5** CB+ neurons increase in the PL throughout life. **a** DCX+ (arrowheads) and CB+ (arrows) cells in the MPL from birth to 67 years. (Inset) High magnification. **b** DCX+ (arrowheads) and CB+ (arrows) cells in the LPL from birth to 77 years. (Right) High magnification. **c–e** Quantification of cells in the LPL from birth to 77 years of age. Each data point corresponds to one individual; line indicates non-linear exponential fit. **c** Density of DAPI+ nuclei in the LPL from birth to 77 years. **d** CB+ cell density in the LPL from birth to 77 years. **e** Percentage of CB+DAPI+ cells in the LPL from birth to 77 years. **f** (Top) 77-year-old amygdala with two LPL cross-sections: one in the BLA containing few DCX+ cells and many CB+ cells and one in the white matter (wm) ventral to the BLA with many DCX+ cells and few CB+ cells. (Inset) higher magnification of the LPL in the white matter. (Bottom) Higher magnification of the LPL in the BLA and examples of small DCX+ cells in this region. **g** CB+TBR1+VGLUT2+ cells (arrows) in the (Left) MPL and (Right) LPL at 13 years. **h** DCX+ and TBR1+ cells in the MPL and LPL at 13 years. (Inset) Double-positive DCX+TBR1+ cell in the MPL. **i** DCX+TBR1+ and TBR1+NeuN+ neurons in the MPL at 13 years. (Top inset) small DCX+TBR1+ cell (arrow) and small DCX+TBR1- cell (arrowhead). (Middle inset) intermediate-sized DCX+TBR1+ cell (arrow) and DCX-TBR1+NeuN+ cell (arrowhead). (Bottom inset) Large DCX-TBR1+NeuN+ cells (arrowheads). **j** DCX+TBR1+VGLUT2- cells (arrows) and DCX-TBR1+VGLUT2+ cells (arrowheads) in the MPL at 13 years. Scale bars: 100 μm (**a**, **b** left, **f** top, **h**), 20 μm (**a** insets, **b** right, **f** top inset, **f** bottom left, **g**, **i** left, **j**), 10 μm (**h** left inset, **i** right insets), 5 μm (**f** lower right insets)

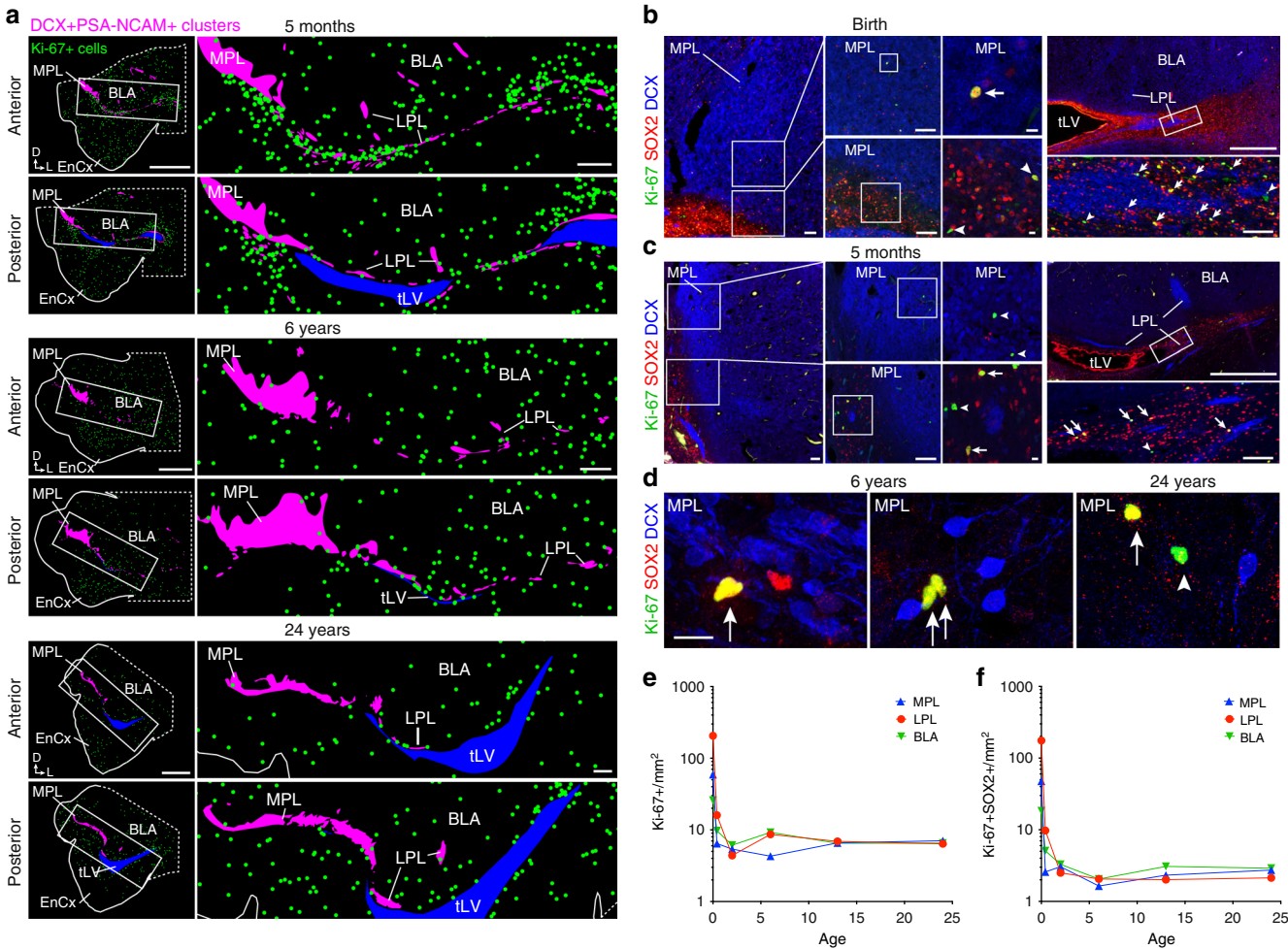

**Fig. 6** Ki-67+ cells in the medial temporal lobe from birth to adulthood. **a** Maps of the location of Ki-67+ cells (green dots) near the DCX+PSA-NCAM+ cell clusters in the PL (purple). Sections of the medial temporal lobe at 5 months, 6 years, and 24 years at anterior and posterior levels separated by 1 mm. (Right) Higher magnification showing location of Ki-67+ cells near the PL and temporal lobe lateral ventricle (tLV). **b–d** Ki-67+SOX2+ cells (arrows) and Ki-67+SOX2- cells (arrowheads) near DCX+ cells in the MPL and LPL at birth (**b**), 5 months (**c**), and 6 and 24 years of age (**d**). **e, f** Quantifications of Ki-67+DAPI+ cells (**e**) and Ki-67+SOX2+ cells (**f**) in the MPL and LPL from birth to 24 years of age in the MPL, LPL, and BLA. Each point represents the average cell density (log-scaled) in each region within one individual at each age. Scale bars: 5 mm (**a** left), 1 mm (**a** right, **b** top right, **c** top right), 100 μm (**b** left, **b** middle left, **b** lower right, **c** left, **c** middle left, **c** lower right), 10 μm (**b** middle right, **c** middle right)

subtypes of excitatory neurons, interneurons, astrocytes, oligo-dendrocytes, microglia", and endothelial cells (Supplementary Fig. 8). The transcripts for DCX, BCL2, and NR2F2 could be detected at much lower expression levels in several other clusters, so we next investigated additional genes we expected to be expressed in the immature PL neurons based on immunostaining. The enzyme responsible for polysialylation of neural cell adhesion

molecule, ST8SIA2 was expressed at the highest level in the immature PL neuron cluster, as were SOX11 and MAP2 but not PROX1 or SP8, consistent with our findings during development (Fig. 8c). The C2 cluster also expressed VGLUT2 (SLC17A6) and at lower levels, TBR1. We confirmed that the DCX+ cells in the PL express the known protein BCL2, as well as the markers of immature neurons SOX11 and ROBO1 (Supplementary Fig. 8).

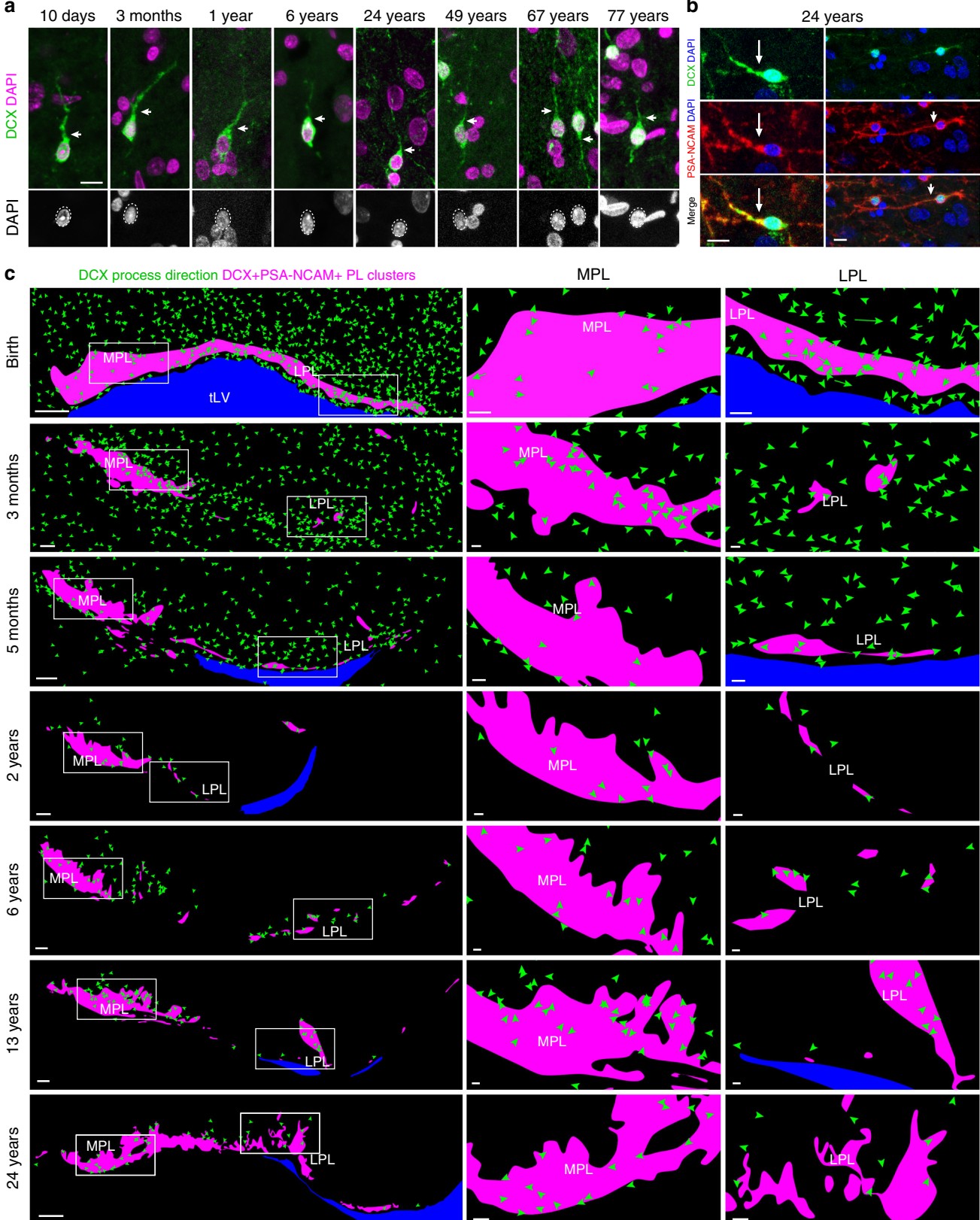

**Fig. 7** DCX+PSA-NCAM+ neurons with migratory characteristics in the postnatal PL. **a** (Top) Examples of DCX+ cells with morphology similar to migratory neurons (arrows) in the PL from 10 days postnatal to 77 years of age. (Bottom) DAPI+ nuclei of the DCX+ cells above showing varying degrees of nuclear elongation (dotted circles). **b** DCX+PSA-NCAM+ cells in the 24-year-old MPL with processes (arrows) and elongated nuclei (left two columns) or rounded nuclei (right column). **c** Maps of the orientations of DCX+ cells with migratory characteristics near the PL from 38 GW to 24 years. (Right) Higher magnification of the MPL and LPL. Arrow direction and length indicate the orientation and length of putative leading processes. Scale bars: 500 μm (**c** left), 100 μm (**c** middle, **c** right), 10 μm (**a**, **b**)

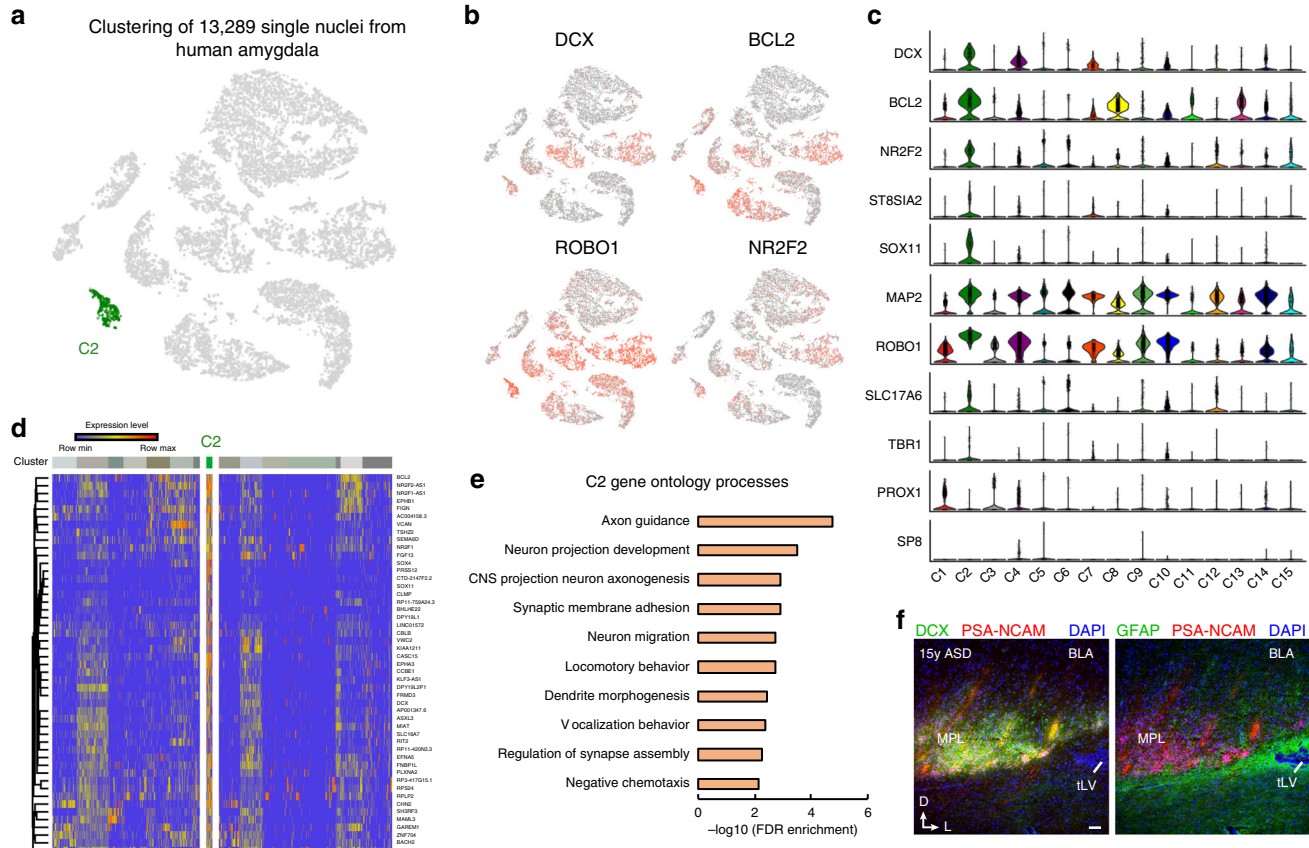

**Fig. 8** Single-nuclei RNA-seq analysis of immature PL cells in the human amygdala. **a** Identification of immature PL neurons based on unbiased clustering (C2). **b**, **c** Expression of markers of immature PL neurons across cell types in the human amygdala. t-SNE plots are colored based on log-transformed and normalized unique molecular identifiers (UMIs). Violin plots are color-coded based on the cluster identity. **d** Unbiased gene signature of immature PL neurons. Expression level corresponds to log-transformed normalized UMIs. **e** Gene Ontology terms enriched for markers genes of immature PL neurons. **f** Example of regions analyzed in the single nuclei RNA-seq dataset: 15-year-old ASD amygdala containing DCX+PSA-NCAM+ regions wrapped with GFAP+ processes, corresponding to the MPL. Scale bar: 100 µm (**f**)

Comparison to other cell types in the amygdala identified a set of genes with enriched expression in the immature PL neurons; this gene set was associated with neuronal projection development and neuronal migration (Fig. 8d, e, Supplementary Data 3). Recent data suggests that PL neurons in the ASD amygdala may not mature at the same rate as in control brains[10]. To look for differentially expressed genes in ASD we compared gene expression in the immature amygdala cell population in age-matched control and ASD brains, after first validating that DCX+PSA-NCAM+ cells were also present in the ASD snRNA-seq tissue sections (Fig. 8f). Comparing nuclei profiles from control and ASD subjects (see the section "Methods"), revealed a small number of ASD-associated differentially expressed genes, many of which are essential for cell adhesion and cytoskeleton function (Supplementary Fig. S8, Supplementary Table 3). Together these results supported a profile of immature neurons that are in progressive stages of maturation and are unique from other cell types in the human amygdala.

## Discussion

Our results show that the human PL is distinct from the CGE during development and contains a population of immature neurons that develop into TBR1+VGLUT2+ excitatory neurons primarily during adolescence, but also in adults. Interestingly, we found a sharp decline in Ki-67+ cells expressing progenitor cell markers in or around the PL after birth;

however, we also found examples of DCX+PSA-NCAM+ cells that could be migratory even in adults. These results cannot rule out the possibility that some of the DCX+PSA-NCAM+ cells continue to be produced in the adult human PL, but indicate that the majority of these neurons are produced before birth or shortly thereafter. Instead of being newly born, the majority of DCX cells in the human PL appear to be in protracted or arrested maturation, maintaining DCX and PSA-NCAM expression, a small size, and simple morphology for decades. This slow maturation, which appears to be an ongoing process throughout life, could be a substrate for protracted plasticity, similar to what is conferred by adult neurogenesis, but independent of progenitor cell divisions.

The amygdala is comprised of an assortment of nuclei with both cortical and striatal features. The principal neurons in the BLA are derived from the lateral and ventral pallium, whereas the interneurons are derived from the medial (MGE) and caudal (CGE) ganglionic eminences[18,44]. The origin of the PL is unknown, but the immature neurons in the PL are located very close to the CGE. Our observation that the PL and CGE both contained high densities of COUP-TFII+ cells during gestation initially appeared to support a relationship between the CGE and PL. However, subsequent analyses revealed that cells in the CGE were also SP8+ PROX1+, and many were Ki-67+ or SCGN+. In contrast to this, the PL was composed mainly of COUP-TFII+SP8−PROX1− cells and few SCGN+ or Ki-67+ cells. This distinction was corroborated by in situ hybridization for TBR1 which showed that

COUP-TFII[+] cells in the PL expressed TBR1, whereas COUP-TFII[+] cells in the CGE did not. Unlike the CGE which generates inhibitory GABAergic interneurons, TBR1 is expressed in pallially derived cortical excitatory cells[40]. Consistently we found that the majority of PL neurons express VGLUT2. The observation that the PL forms distinctly from the CGE supports the conclusion that this region contains immature excitatory neurons, and suggests that the PL originates from the lateral or ventral pallium.

The possibility of adult amygdala neurogenesis has been investigated in animal models. In rodents, DCX[+]PSA-NCAM[+] cells have been observed in layer 2–3 of the piriform cortex, but there is no discernable PL in rodents and the piriform cortex DCX[+] cells are not born in adulthood[24,26]. In contrast, several studies using the thymidine analog BrdU birthdating and co-localization with neuronal markers concluded that there is limited persistent neurogenesis in the rodent amygdala and piriform cortex[1,2]. Similarly, in two non-human primates (*Saimiri sciureus* and *Macaca fascicularis*), BrdU-labeled neurons were detected in the amygdala PL and piriform cortex[3]. Confirmation of adult neurogenesis remains highly controversial in other parts of the brain where these tools have been used alone[45]. Investigation of thymidine analog-labeled cells using robust standards like electron microscopy, physiology, and connectivity[46–49], to confirm that the new cells are adult-born neurons has not been applied to the adult amygdala. Our findings in the human cannot completely distinguish between slow maturation of neurons and newly formed cells; however, the adult PL is clearly a unique region in the human brain where it is frequent to find immature neurons throughout life. This is in sharp contrast to the adult human dentate gyrus, where immature neurons become rare early in adolescence[50].

Our results demonstrate that in the PL there is a continued presence of immature neurons in adolescents and adults, even in individuals 77 years old. However, the number of dividing (Ki-67[+]) cells declined rapidly in the PL and neighboring regions within the first year of life. The number of Ki-67[+] cells at 2 years of age was similar to that observed in adults. In ages 2 years and older, Ki-67[+] cells in the PL and around the PL were primarily in blood vessels or in oligodendrocyte (OLIG2[+]) cells. Interestingly a subpopulation of Ki-67[+] cells expressed a marker associated with neural stem cells, SOX2 (Fig. 6b–f); however, the Ki-67[+] cells in the adult were BLBP[−] or vimentin[−] (Supplementary Figs. 6 and 7). The density of the Ki-67[+]SOX2[+] cells was similar between the MPL, LPL, and the nearby BLA, suggesting that the PL is not unique in these proliferative cell types. This suggests that neurogenesis in the PL comes to an end early in life, but cannot rule out that Ki-67[+] SOX2[+] cells could be producing new neurons in this region. We did observe Ki-67[+]SOX2[+] cells in close proximity (5–20 μm) to DCX[+] cells with migratory features at 6 and 24 years (Fig. 6d). Some of the remaining BLBP[+] or vimentin[+] cells (Ki67[−]) in the PL had long radial processes, even in adults (Supplementary Fig. 6). It is possible that under some conditions like injury, quiescent stem cells become re-activated and produce new neurons for the PL, as has been suggested following uncal hippocampal lesions in the monkey[34].

The low number of Ki-67[+] progenitors in the PL and adjoining regions, combined with our observation that this region contains dense DCX[+]PSA-NCAM[+] cells at birth, suggests that the majority of neurons within the PL are slowly maturing. This slow maturation of DCX-retaining TBR1[+] neurons has been recently shown for the layer II piriform cortex cells in mice where lineage tracing is possible[25]. Similar populations of DCX-expressing cells have been described in adult sheep, whales, and dolphins[51,52] in several regions outside of layer II of the cortex, including the PL[22]. Work in sheep using BrdU indicates that cells that retain DCX into adulthood are born during fetal development[24]. Our finding that the number of NeuN[+] and CB[+] neurons increases in adolescence and adulthood within the PL also support the conclusion that in humans the DCX[+]PSA-NCAM[+] cells are slowly maturing during postnatal life. It is also possible that a sub-population of the DCX[+]PSA-NCAM[+] cells in the PL are migrating away or dying, contributing to their decline with increasing age. The presence of DCX[+]PSA-NCAM[+] cells with an elongated morphology similar to that of migrating young neurons in the adult (up to 77 years of age) supports the possibility that these neurons continue adjusting their final position. Our maps of locations and orientations of putative young migrating neurons do not support a uniform directional migration of cells into or away from the human PL (Fig. 7c). Instead many of these elongated cells appear to be randomly oriented, which would predict contributions to the PL or other very close-by regions. For example, some immature neurons in the PL may move to the nearby basal and accessory basal nuclei of the amygdala to account for postnatal increases in neuron numbers reported there[10]. Our results also suggest that it is unlikely that cells are recruited into the PL from a distant location via long-range neuronal migration, as most of the DCX[+]PSA-NCAM[+] elongated cells after 2 years of age are found within or immediately adjacent to the PL (Fig. 7c). It is possible that some of these cells retain short-range migratory behavior for years to adjust the final location of their soma. In songbirds, young neurons continue to adjust the location of their cell body in the last stages of neuronal migration in a wandering behavior that maybe key to their ultimate integration[53].

The results from single nuclei sequencing confirm that cells in the PL have an immature transcriptional profile. The functions of the top differentially expressed genes in this population relative to other amygdala regions were related to developmental processes, such as axon growth and neuron migration. To capture enough nuclei for snRNA-seq analysis, our analysis focused on young ages (4–15 years), when there are still many immature neurons present and the greatest changes in their maturation were occurring. Impaired survival, migration, or maturation of the immature PL neurons could underlie some of the postnatal changes in total neuron number in the PL and BA that have been recently detected in ASD[10]. For example, the gene LRP1B which suppresses cell growth and motility in tumor cells by regulating focal adhesions and the actin cytoskeleton[54] is expressed at higher levels in ASD immature neurons compared to controls (Supplementary Fig. 8). The identification of genes expressed in the immature PL neurons in the human might facilitate the identification of similar cells in other species.

The most substantial changes in the composition of the PL occurred during adolescence. The medial nucleus (MPL) had a sharp drop in DCX[+]PSA-NCAM[+] cells between 6 and 24 years, and in adults was much less prominent than in children (Fig. 3a). In contrast, small clusters of dense LPL with percentages of DCX[+]PSA-NCAM[+] cells (similar to LPL at birth) could still be observed at 77 years of age next to other LPL clusters with only 1–2 DCX[+] cells per-section (Fig. 5f). This suggests that there may be a functional significance in the differential recruitment of individual PL neurons into amygdala circuits. Tracer studies[6] in the macaque have found stripes of cells in the uncus of the hippocampus that have afferent projections preferentially into the MPL. Interestingly, the MPL matures earlier than the lateral PL. The uncus of the hippocampus is important for autobiographical memory, suggesting a possible role for the PL in processing contextual inputs. Efferents from the PL include the entorhinal cortex[55], BLA[56], and the central amygdala, which may suggest that the PL is an important relay of information from the uncal hippocampus to amygdala outputs, however it is not known if cells maturing in adults share this connectivity. If the protracted

development in the PL has a function similar to that of adult neurogenesis in the dentate gyrus or olfactory bulb[33,57], the PL could be a pattern separator for emotional contextual information relayed from the hippocampus to the amygdala.

This study shows a substantial change in the structure of hippocampal–amygdala circuitry that occurs during human childhood and adolescence. The cellular architecture and the maturation of PL neurons could therefore be influenced by life experiences during these ages. A recent study found that in the macaque, early life maternal separation leads to lower levels of TBR1 gene expression in the PL later in life[35]. Which cells are affected in the PL and whether this is merely a correlation, or reflects lower amounts of activity in the PL or lower levels of cell maturation remains to be tested. Our data indicate that human amygdala circuits continue to undergo unique changes in neuron shape, size, and maturation during essential formative years in childhood and later in life. Perhaps most strikingly, this process does not appear to rely on the birth of new neurons exclusively, but rather the maintenance of populations of immature excitatory neurons in a juvenile state for protracted periods of postnatal life. The manner in which this process is regulated and the types of genetic and environmental influences that can affect it are exciting directions for future investigation.

## Methods

**Human tissue collection**. Forty-nine post-mortem human amygdala specimens were collected for this study (Supplementary Table 1). The authors confirm that this study complies with all relevant ethical regulations for work with human samples and that informed consent was obtained. Tissue was collected with previous patient consent following institutional ethical regulations: 1. The University of California, San Francisco (UCSF) Committee on Human Research. Protocols were approved by the Human Gamete, Embryo and Stem Cell Research Committee (Institutional Review Board) at UCSF. 2. The Ethical Committee for Biomedical Investigation, Hospital la Fe (2015/0447) and the University of Valencia Ethical Commission for Human Investigation. 3. The study was performed according to the Declaration of Helsinki and approved by the Ethical Committee on Clinical Research of the University Hospital of Albacete, on its meeting of January 2015. For infant cases, when the brain is at full term (37–40 gestational weeks) and autopsy performed within 2 days after birth, we refer to this as birth. We collected tissue blocks from the temporal lobe, beginning at the anterior dentate gyrus and ending at the anterior tip of the temporal lobe. All brains were cut into ~1.5 cm blocks, fixed in 4% PFA for an additional 2 days, cryoprotected in a 30% sucrose solution, and then frozen in OCT (Sakura Finetek). Blocks were cut into 20 μm sections on a cryostat and mounted on SuperFrost Plus (Fisher) slides for immunohistochemistry.

**Immunohistochemistry**. Frozen slides were allowed to equilibrate to room temperature. Some antigens required antigen retrieval (Supplementary Table 2), which was conducted at 95 °C in 10 mM Na citrate buffer, pH = 6.0 in a BioWave microwave oven (Pelco). Following antigen retrieval, slides were washed with PBS-T buffer (0.05% Triton X-100 in PBS) for 10 min, placed in 1% $H_2O_2$ in PBS for 45 min and then blocked with TNB solution (0.1 M Tris–HCl, pH 7.5, 0.15 M NaCl, 0.5% PerkinElmer TSA blocking reagent) for 1 h. Slides were incubated in primary antibodies overnight at 4 °C (Supplementary Table 2) in TNB. The next day, slides were washed with PBS-T (3 × 10 min), then incubated in biotinylated (Jackson Immunoresearch Laboratories) or Alexa (Invitrogen) secondary antibodies for 2.5 h at room temperature in TNB. Slides were incubated for 30 min in streptavidin–horseradish peroxidase diluted (1:200) with TNB, were washed, and then were incubated in tyramide-conjugated fluorophores for 5 min diluted in amplification diluent:fluorescein: 1:50; Cy3: 1:100; Cy5: 1:100. Staining was conducted in technical triplicate prior to analysis. Conditions of use for each antibody was validated by the manufacturer. We evaluated each antibody by comparison to no primary antibody (negative) controls, comparison to (often enriched) expression in human gestational tissue, and comparison to rodent staining patterns. Antibodies and dilutions used for this study are listed in Supplementary Table 2 and here: ALDH1L1 (1:500, NeuroMab N103/39), BCL2 (1:100, Santa Cruz Biotech sc-7382 (C-2)), BLBP (1:200, EMD Millipore ABN14), BLBP (1:200, Abcam ab131137), Calbindin (1:1000, Swant CB-38a), Calretinin (1:1000, Swant 6B3), COUP-TFII (1:250, R&D Systems PP-H7147–00), Doublecortin (1:200, Cell Signaling 4604S), Doublecortin (1:200, EMD Millipore AB2253), GFAP (1:750, Abcam ab4674), Iba1 (1:100, Wako, 019-1974), Ki-67 (1:200, BD Pharmigen 556003), Ki-67 (1:500, Novocastra NCL-Ki67p), Ki-67 (1:1000, Vector Labs VP-K451), MAP2 (1:500, Abcam ab5392), Nestin (1:250, Covance MMS-570p), NeuN (1:500, EMD Millipore ABN91), NeuN (1:1000, Novus Biologicals R-3770-100), Neurofilament (1:1000, Abcam ab24574), NKX2.1 (1:500, Santa Cruz Biotech sc-

13040 (H-190)), nNOS (1:500, EMD Millipore AB5380), NPY (1:500, Abcam ab30914), OLIG2 (1:750, EMD Millipore AB9610), PROX1 (1:500, EMD Millipore AB5475), PROX1 (1:500, R&D Systems AF2727), PSA-NCAM (1:1000, EMD Millipore MAB5324), ROBO1 (1:100, Santa Cruz Biotech sc-293444), SCGN (1:1000, Sigma-Aldrich HPA006641), SOX2 (1:200, Santa Cruz Biotech sc-17320 (Y-17)), SOX2 (1:200, Cell Signaling 2748S), SOX11 (1:500, EMD Millipore AB5776), SP8 (1:200, Santa Cruz Biotech sc-104661 (C-18)), SST (1:250, Santa Cruz Biotech sc-7819 (D-20)), TBR1 (1:200, EMD Millipore AB2261), TUJ1 (1:200, Covance MMS-435P), VGLUT2 (1:200, EMD Millipore AB5907) Vimentin (1:1000, Sigma-Aldrich V5255)

**Fluorescent microscopy, image processing, and quantifications**. Images were acquired on Leica TCS SP8 or SP5 confocal microscopes using ×10/0.3 NA, ×20/0.7 NA, or ×63/1.4 NA objectives. Tile-scans were acquired on a Zeiss Axiovert 200M microscope with either ×10/0.3 NA or ×20/0.45 NA objectives. Images were analyzed using Neurolucida (MBF Bioscience 2017) and ImageJ. For quantifications, ×63 confocal z-stacks were collected from three to five evenly spaced sections across the region at each age. The PL within each section was sampled exhaustively. Cell counting was conducted blinded to identifying case information. Linear adjustments to image brightness and contrast were made in Adobe Photoshop (CS6) and figures were assembled in Adobe Illustrator (CS6).

**Statistics**. Statistical analyses were performed using R (R Core Team 2016) and lme4. A linear mixed effects analysis was conducted on the relationship between age and cell populations (DAPI+, DCX+PSA-NCAM+, NeuN+, CB+, Ki-67+) in the amygdala. Age and region were considered as fixed effects. Random effects had intercepts for individual and section analyzed, and by-individual and by-section random slopes for the effect of age. The dependent variables were log-transformed to ensure that the data and residuals had a normal distribution and no evident deviations from homoscedasticity. Likelihood ratio tests (LRTs) of the full model and the model without the effect of interest were used to evaluate significance. Graphs and one-way ANOVA tests were generated in GraphPad Prism (6.0).

**RNA scope in situ hybridization**. Sequences of target probes, preamplifier, amplifier, and label probe are proprietary and commercially available (Advanced Cell Diagnostics, Hayward, CA). Typically, the probes contain 20 ZZ probe pairs (~50 bp/pair) covering ~1000 bp. Here, we used a probe against human TBR1 targeting NM_006593.2 as a single-plex probe. Slides for ISH were initially taken from −80 °C, dried at 60 °C for 1 h, and fixed in 4% PFA for 2 h. After three PBS washes, slides were treated with ACD hydrogen peroxide for 10 min and then washed in water 2x before treatment in 1x target retrieval buffer (ACD) for 5 min (at 95–100 °C). After washing in water and then 100% alcohol, the slides were left to dry overnight before protease treatment for 15 min at 40 °C in the RNAscope oven. Hybridization of probes and amplification solutions was performed according to the manufacturer's instructions. In short, tissue sections were incubated in desired probe (~2–3 drops/section) for 2 h at 40 °C. The slides were washed two times in 1x wash buffer (ACD) for 2 min each. Amplification and detection steps were performed using the RNAscope 2.5 HD Red Detection Kit reagents (ACD, 320497) for single-plex probes. Sections were incubated with Amp1 for 30 min at 40 °C and then washed two times in wash buffer for 2 min each. Amp2 was incubated on the sections for 15 min at 40 °C, followed by two washes in wash buffer. Sections were incubated in Amp3 for 30 min at 40 °C and washed two times in wash buffer for 2 min each, followed by incubation of Amp4 for 15 min at 40 °C. Slides were washed two times in wash buffer for 2 min each. Slides were incubated with Amp5 for 30 min at RT using the HybEZ humidity control tray and slide rack to maintain humidity. The slides were washed two times in 1x wash buffer for 2 min each and incubated in Amp6 for 15 min at RT before washing two times in wash buffer for 2 min each. ISH signal was detected by diluting Fast RED-B in Fast RED-A solution (1:60 ratio) and incubating sections in this solution for 10 min. Slides were washed in water two times to stop the reaction.

**Processing of brain tissue samples for snRNA-seq**. De-identified snap-frozen post-mortem tissue samples from ASD patients and controls were obtained from the University of Maryland Brain Bank through the NIH NeuroBioBank. Samples were sectioned on a cryostat to collect 100 μm sections for total RNA isolation and nuclei isolation. Total RNA from ~10 mg of collected tissue was isolated and used to perform RNA integrity analysis on the Agilent 2100 Bioanalyzer using RNA Pico Chip assay. Only samples with RNA integrity number (RIN) >6.5 were used to perform nuclei isolation and single-nucleus RNA sequencing (snRNA-seq).

**Nuclei isolation and snRNA-seq on the 10x Genomics platform**. Matched control and ASD samples were processed in the same nuclei isolation batch to minimize potential batch effects. 40 mg of sectioned brain tissue was homogenized in 5 mL of RNAase-free lysis buffer[58] (0.32 M sucrose, 5 mM $CaCl_2$, 3 mM $MgAc_2$, 0.1 mM EDTA, 10 mM Tris–HCl, 1 mM DTT, 0.1% Triton X-100 in DEPC-treated water) using glass dounce homogenizer (Thomas Scientific, Cat # 3431D76) on ice. The homogenate was loaded into a 30 mL-thick polycarbonate ultracentrifuge tube (Beckman Coulter, Cat # 355631). 9 mL of sucrose solution (1.8 M sucrose, 3 mM

MgAc$_2$, 1 mM DTT, 10 mM Tris–HCl in DEPC-treated water) was added to the bottom of the tube with the homogenate and centrifuged at 107,000 × $g$ for 2.5 h at 4 °C. Supernatant containing cell debris, organelles rather than nuclei, and free-floating RNA and protein was aspirated, and the nuclei pellet was incubated in 250 μL of DEPC-treated water-based PBS for 20 min on ice before resuspending the pellet. Resuspended nuclei were filtered twice through a 30 μm strainer. Nuclei were counted using a hemocytometer and diluted to 2000 nuclei/μL before performing single-nuclei capture on the 10x Genomics Single-Cell 3′ system. Target capture of 3000 nuclei per sample was used; the 10x capture and library preparation protocol was used without modification. Single-nuclei libraries from individual samples were pulled and sequenced either on the HiSeq 2500 machine.

**snRNA-seq data processing and filtering**. For library demultiplexing, fastq file generation and read alignment and UMI quantification, CellRanger software v 1.3.1 was used. CellRanger was used with default parameters, except for using pre-mRNA reference file (ENSEMBL GRCh38) to insure capturing intronic reads originating from pre-mRNA transcripts abundant in the nuclear fraction. Individual expression matrices containing numbers of unique molecular identifiers (UMIs) per nucleus per gene were filtered to retain nuclei with at least 400 genes expressed and <10% of total UMIs originating from mitochondrial and ribosomal RNAs. Genes expressed in less than three nuclei were filtered out. Mitochondrial RNA genes were filtered out as well to exclude transcripts coming from outside the nucleus to avoid biases introduced by nuclei isolation and ultracentrifugation. Individual matrices were combined, UMIs were normalized to the total UMIs per nucleus and log transformed.

**Dimensionality reduction, clustering, and t-SNE visualization**. Since two sequencing platforms were utilized to generate the data, we sequenced four libraries on both platforms and identified genes associated with sequencing platform using MAST[59] (FDR < 0.05; FC > 0.1), which were then filtered out. Filtered (containing genes expressed in more than five cells) log-transformed UMI matrix was used to perform truncated singular value decomposition (SVD) with $k = 50$. Scree plot was generated to select the number of significant principle components (PCs) by localizing the last PC before the explained variance reaches plateau. The significant PCs were used to calculate Jaccard distance-weighted nearest-neighbor distances; number of nearest neighbors was assigned to root square of number of nuclei. Resulting graph with Jaccard-weighted edges was used to perform Louvain clustering[60]. To visualize nuclei transcriptomic profiles in two-dimensional space, t-distributed stochastic neighbor embedding (t-SNE)[61] was performed with the selected PCs and combined with cluster annotations.

**Cell type annotation and quantification**. Cell types were annotated based on expression of known marker genes visualized on the t-SNE plot and by performing unbiased gene marker analysis. For the latter, MAST was used to perform differential gene expression analysis by comparing nuclei in each cluster to the rest of nuclei profiles. Genes with FDR < 0.05 and log-fold change of one or more were selected as cell-type markers. For heatmap generation and visualization, Morpheus was used (https://software.broadinstitute.org/morpheus).

**Differential gene expression analysis**. To identify genes differentially expressed in ASD compared to control in each cell type, MAST was used to perform zero-inflated regression analysis by fitting a linear mixed model (LMM). LMM included age, sex, RIN, and post-mortem interval. We accounted for the fact that multiple nuclei were captured from each individual using a hierarchical model design. The following model was fit with MAST:

zlm(~diagnosis + (1|ind) + cngeneson + age + sex + RIN + PMI, sca, method = "glmer", ebayes = F, silent = T)

where cngeneson is gene detection rate (factor recommended in MAST tutorial), ind is individual label, RIN is RNA integrity number and PMI is post-mortem interval. To identify genes differentially expressed due to the disease effect, LRT was performed by comparing the model with and without the diagnosis factor. Genes with fold change of expression of at least 0.14 (10% difference) and FDR < 0.05 were selected as differentially expressed. In addition, we calculated raw fold change of gene expression by running MAST with only the diagnosis factor in the model and filtered genes with raw fold change <10%.

**Statistical overrepresentation test for Gene Ontology terms**. PANTHER[62] was used to perform statistical overrepresentation test for DEGs from each cluster. All genes tested for differential expression in a given cluster were used as the background and GO Biological Processes ontology was used. Processes with FDR < 0.05 were considered and sorted by FDR.

**Reporting summary**. Further information on research design is available in the Nature Research Reporting Summary linked to this article.

## Data availability

Raw RNA-seq data (fastq files) are available at Sequence Read Archive (SRA), accession number PRJNA530977. The authors declare that the data supporting the findings of this study are available within the paper and its supplementary information files.

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

## Acknowledgements

S.F.S. was supported by F32 MH103003 and M.F.P. was supported by K08 NS091537. S.M. was supported by a postdoctoral fellowship from the German Research Foundation (DFG, MA 7374/1-1). A.A.-B. was supported by NIH grants P01 NS083513, R01 NS028478, and a gift from the John G. Bowes Research Fund. He is the Heather and Melanie Muss Endowed Chair and Professor of Neurological Surgery at UCSF. J.L.R. was supported by NINDS grant R01 NS099099. We gratefully acknowledge technical assistance provided by Cristina Guinto, Jose Rodriguez, Joseph Elsbernd, and Mike Wong, and discussions with Kevin Chu and Justin Ellis. This work would not be possible without the generous anatomical donations of all of the individuals included in this study.

## Author contributions

The study was conceived and designed by S.F.S. and A.A.-B. Staining was performed by S.F.S., V.H.-P., S.M., M.F.P and K.S. Image and data collection was conducted by S.F.S. Amygdala sn-Seq experiments were conducted by D.V. Ultrastructural analysis was performed by V.H.-P. and J.M.G.-V. Additional tissue samples and neuroanatomy were provided by E.F.C. and R.I. Data analysis and interpretation was conducted by S.F.S., M. F.P., V.H.-P., A.R.K., J.L.R, J.M.G.-V., E.J.H. and A.A.-B. S.F.S. and A.A.-B. wrote the manuscript with input from all authors.

## Additional information

**Competing interests:** A.R.K., J.L.R., and A.A.-B. are co-founders and serve on the scientific advisory board of Neurona Therapeutics. The remaining authors declare no competing interests.

