## [Peer Review File · Nature Communications]

REVIEWERS' COMMENTS:

Reviewer #1 (Remarks to the Author):

The authors have addressed my major concerns from the original submission. The new statistical analyses, additional quantifications, and the RNA-seq strengthen this paper. I applaud the authors for the painstaking analyses and beautiful images in this paper.

Reviewer #2 (Remarks to the Author):

My apologies to the authors for the very long delay in my response. In the revised manuscript, the authors have addressed most of the concerns that I had raised. I am well aware that functional studies are not really feasible in humans. The apparent lack of a PL region in rodents makes it doubly difficult. They have provided new data suggesting that these late maturing neurons in the human PL may be integrated into. Overall, this is a very interesting study and, quite apart from a description of this interesting region, it raises issues about direct comparisons between neural circuits that mediate emotional responses in humans and rodent models, particularly the changes that take place during adolescence. I have no further concerns.

Pankaj Sah

Reviewer #3 (Remarks to the Author):

This paper examines evidence for the location, cellular phenotype, development, and fate of immature neurons in the PL from mid-gestation through advanced adulthood. These studies are a novel, and possibly paradigm-shifting, set of data that inform that way the amygdala development in primates is conceptualized. The paralaminar region is a dynamic region in humans (and in larger, longer-lived species) that suggests that there is an adaptation in these mammals for amygdala growth over the life span. A copious amount of work has been done. Some of the details are excessive, and in this reviewer's mind detract at times from the main points and focus of the paper. Notwithstanding, among the 'take-homes', the most important points seem to be:

-The PL progenitors appear not to derive from the caudal ganglionic eminence, which produces interneurons. Markers of CGE such as COUP-II also define PL progenitors early on, but PL progenitors have a different cell fate and position in the amygdala.

-The PL is a distinct region of immature neurons in the human which is actively dividing through the 2nd year of life and then are largely finished with divisions.

-The PL post-mitotic neurons are largely destined to become glutamatergic, and that mature PL neurons themselves are glutamatergic.

-Maturation of immature to mature neurons begins in adolescence, but is maintained during the lifespan.

-The ratio of immature to mature neurons decreases, first in the MPL and later in the LPL.

-Gap junctions characterize communication among the immature neurons.

The paper's strength, in addition to the size of their brain bank and preservation of the sample, is the authors' expertise in brain development and immunostaining techniques. There have been a number of control experiments to dissect the origination of the PL (which is not from the CGE as initially supposed, due to its proximity). How progenitors from the pallium arrive to the ventral amygdala is still a mystery, but it is clear that the CGE is not the source of these neurons. Understanding the timing of proliferation, and then progression from post-mitotic precursor to mature VGLUT2 positive neuron is also a fundamental contribution to the literature.

A host of other studies including EM, RNA-seq, and examination of ASD brain are valuable, but will need to be fleshed out further in other studies. This study is most important as an advance in understanding the protracted development of brains in large mammals with long lifespans. Scientific views have been too long been biased towards findings in smaller species. In translational neuroscience, we have made slow progress in understanding development and plasticity of limbic system structures in humans due to over-reliance on animal models that do not anatomically approximate the human. This has led to few treatments or preventative strategies for psychiatric disorders (virtually all of which have functional abnormalities in the amygdala). This paper helps shed light on a very novel cellular region in the amygdala that can only be elucidated through post-mortem studies in the human. While there will be criticisms that the study is not 'mechanistic', there are no rodent models that will allow interrogation of this specific cellular group which is unique and likely functionally important.

I have some minor criticisms that mainly have to do with the writing and organization.

Introduction

Bottom of page 2: first use of DCX and PSA-NCAM. Please spell out, and more importantly, give putative function of the proteins, and known expression in other post-mitotic cells.

Bottom of page 3: same issue for COUP1 and TBR1. Need to directly state the significance of these proteins for cell identity and function in introduction.

Results

Writing:

Page 4: 'Regions of with high Ki67+ cell density contained many SP8+COUP1+ cells, few NKK2.1+ cells.....' I am following distinguishing the PL from CGE because you have given your criteria, but what are NKK2.1+ cells part of?

TBR1-significance should be mentioned earlier, in this case, in Introduction. The reader still isn't sure why this is important, except the implication that it is not found in CGE.

'We next studied cells in late gestation and early postnatal life.' Why? The reader can assume why, but please state why you chose this interval.

Did all DCX+ cells co-express TBR1, and if not what percent of the DCX population co-expressed it at this age?

Page 5: What does the area under the clusters tell us at each age? It is hard to know if this is due to soma size, packing density, # cells, neuropil. The information that follows is more informative (ratios of NeuN+ cells to DCX-PSANCAM+ cells).

The 3D map of the DCX-PSA-NCAM clusters is interesting. Did you differentiate the PL extensions from Intercalated islands at these ages?

Page 8: RNA-seq data is interesting, although the single-nucleus technique is not well-explained in the methods. How are intact nuclei separated from free mRNA (cytosolic) in the cell? What is the significance of SOX11 and ROBO1? Are they confirmed to be nuclear mRNA and is their protein product known? Why was a 10-fold threshold chosen? This seems low-how is it related to levels detectable with in situ hybridization?

The temporal dissociation between the mPL and IPL maturation is very interesting, particularly given that the hippocampus projection is especially dense to the mPL vs. IPL. The implication is that the hippocampal afferents might have their greatest impact on neural maturation in adolescence.

Response to reviewers:

Reviewer #1 (Remarks to the Author):

The authors have addressed my major concerns from the original submission. The new statistical analyses, additional quantifications, and the RNA-seq strengthen this paper. I applaud the authors for the painstaking analyses and beautiful images in this paper.

We thank this reviewer for their helpful comments in preparing our manuscript.

Reviewer #2 (Remarks to the Author):

My apologies to the authors for the very long delay in my response. In the revised manuscript, the authors have addressed most of the concerns that I had raised. I am well aware that functional studies are not really feasible in humans. The apparent lack of a PL region in rodents makes it doubly difficult. They have provided new data suggesting that these late maturing neurons in the human PL may be integrated into. Overall, this is a very interesting study and, quite apart from a description of this interesting region, it raises issues about direct comparisons between neural circuits that mediate emotional responses in humans and rodent models, particularly the changes that take place during adolescence. I have no further concerns.

We appreciate this author's comments on our original submission and thank them for their suggestions.

Reviewer #3 (Remarks to the Author):

I have some minor criticisms that mainly have to do with the writing and organization.

Introduction

Bottom of page 2: first use of DCX and PSA-NCAM. Please spell out, and more importantly, give putative function of the proteins, and known expression in other post-mitotic cells.

We have modified the text to address this point in paragraph 2:

"The primate amygdala does, however, contain a large population of immature neurons expressing the immature markers doublecortin (DCX) and polysialylated neural cell adhesion molecule (PSA-NCAM) in a region called the paralaminar nuclei (PL)." In paragraph 4 we now emphasize their known expression in other regions and species.

Bottom of page 3: same issue for COUP2 and TBR1. Need to directly state the significance of these proteins for cell identity and function in introduction.

We have modified the introduction to explain the transcription factors used in the study as well as their significance for cell identity.

Results

Writing:

Page 4: 'Regions of with high Ki67+ cell density contained many SP8+COUP2+ cells, few NKK2.1+ cells..... ' I am following distinguishing the PL from CGE because you have given your criteria, but what are NKK2.1+ cells part of?

We have clarified that NKK2.1 expression would indicate that these cells are likely derived from the medial ganglionic eminence lineage.

TBR1-significance should be mentioned earlier, in this case, in Introduction. The reader still isn't sure why this is important, except the implication that it is not found in CGE.

We have modified the text to address this point; thank you for this suggestion.

'We next studied cells in late gestation and early postnatal life.' Why? The reader can assume why, but please state why you chose this interval.

We have modified the text to clarify this “We next asked if the PL contained similar cellular identities at older ages studied the cells in the PL in by studying the ventral amygdala in late gestation and early postnatal life.”

Did all DCX+ cells co-express TBR1, and if not what percent of the DCX population co-expressed it at this age?

At birth the expression of TBR1 is just appearing in PL neurons and quantifications were not possible, however the mRNA was present in nearly all of the DCX cells in the PL at this age (Supplementary Figure 1c,d). We have quantified the TBR1/DCX cells in the 13 year old sample “Of the small DCX⁺ cells, 54.4 % (± 3.4 %, SD; CI: 50.22 – 58.56; 972 TBR1⁺DCX⁺ / 1762 DCX⁺ cells) were TBR1⁺”.

Page 5: What does the area under the clusters tell us at each age? It is hard to know if this is due to soma size, packing density, # cells, neuropil. The information that follows is more informative (ratios of NeuN+ cells to DCX-PSANCAM+ cells).

We feel that it is helpful to know that while the ratios of the different cell types are changing, the overall size of the paralamina nuclei is increasing considerably (not decreasing or staying the same). Our data suggest that this is likely due to the increase in NeuN+ cell soma size and the increase in the neuropil of the PL neurons.

The 3D map of the DCX-PSA-NCAM clusters is interesting. Did you differentiate the PL extensions from Intercalated islands at these ages?

We have modified the text and figure legend to make it clear what our maps show. In figure 2 (at Birth) the intercalated nuclei are outside the mapped area; the mapped DCX/COUPTFII clusters in this figure correspond to the PL. In Supplementary Figure 3 (at 15 years) the sagittal 3D reconstructions contain the PL and the intercalated islands, both of which express DCX and PSA-NCAM. We have indicated with an arrow the intercalated nuclei.

Page 8: RNA-seq data is interesting, although the single-nucleus technique is not well-explained in the methods. How are intact nuclei separated from free mRNA (cytosolic) in the cell? What is the significance of SOX11 and ROBO1? Are they confirmed to be nuclear mRNA and is their protein product known? Why was a 10-fold threshold chosen? This seems low-how is it related to levels detectable with in situ hybridization?

We apologize for not explaining the nuclei isolation and snRNA-seq methods in more detail. We modified the Methods section “Nuclei isolation and snRNA-seq on the 10x Genomics platform” to clarify the nuclei isolation procedure. In our protocol, nuclei are isolated by ultracentrifugation through a sucrose cushion. Therefore, cell debris, organelles rather than nuclei and soluble material, including free-floating RNA and protein are removed during centrifugation. Only nuclei pelleted during centrifugation are resuspended and used during capture on the 10x Genomics platform. In addition, we perform filtration of nuclei suspension through a 30 um strainer twice.

SOX11 and ROBO1 were identified as genes whose expression was enriched in immature neuronal population (C2) compared to other cell types in the amygdala captured with snRNA-seq. This is referred to as “unbiased

gene marker analysis” in the “Cell type annotation and quantification of regional and individual contribution to cell types” Methods section.

For our snRNA-seq data, most of the reads are coming from unspliced (pre-mRNA) transcripts. Therefore, SOX11 and ROBO1 transcripts we detected are most likely pre-mRNA transcripts enriched in the nucleus. We confirmed protein co-expression of ROBO1 and SOX11 protein with DCX (marker of immature neurons) in Supplementary Figure 8.

For 10-fold threshold, we assume the reviewer means 10% difference of expression between Control and ASD that was chosen as the threshold. For most genes differentially expressed between Control and ASD, fold changes were observed were small, which is consistent with bulk RNA-seq data from ASD patient samples. However, we should note that for SOX11 and ROBO1, these genes were identified as markers of C2 cluster (immature neurons), rather than differentially expressed in ASD. These genes are highly specific to C2, with ROBO1 expressed on a level that is 14 times higher in C2 than in other cells in the amygdala, and SOX11 expressed at 2 times higher level. We did not perform in situ validation of ROBO1 and SOX11 since we validated expression of these genes on the protein level (Supplementary Figure 8).

The temporal dissociation between the mPL and IPL maturation is very interesting, particularly given that the hippocampus projection is especially dense to the mPL vs. IPL. The implication is that the hippocampal afferents might have their greatest impact on neural maturation in adolescence.

We agree that this might have an interesting functional significance, and we now highlight this point and the macaque studies that have shown preferential projections from the hippocampus to the MPL in the discussion.

--

We would like to thank all of the reviewers for their time and helpful suggestions.